

# Compact graphical representation of phylogenetic data and metadata with GraPhlAn

Francesco Asnicar[1], George Weingart[2], Timothy L. Tickle[3], Curtis Huttenhower[2,3] and Nicola Segata[1]

[1] Centre for Integrative Biology (CIBIO), University of Trento, Italy
[2] Biostatistics Department, Harvard School of Public Health, USA
[3] Broad Institute of MIT and Harvard, USA

## ABSTRACT

The increased availability of genomic and metagenomic data poses challenges at multiple analysis levels, including visualization of very large-scale microbial and microbial community data paired with rich metadata. We developed GraPhlAn (Graphical Phylogenetic Analysis), a computational tool that produces high-quality, compact visualizations of microbial genomes and metagenomes. This includes phylogenies spanning up to thousands of taxa, annotated with metadata ranging from microbial community abundances to microbial physiology or host and environmental phenotypes. GraPhlAn has been developed as an open-source command-driven tool in order to be easily integrated into complex, publication-quality bioinformatics pipelines. It can be executed either locally or through an online Galaxy web application. We present several examples including taxonomic and phylogenetic visualization of microbial communities, metabolic functions, and biomarker discovery that illustrate GraPhlAn's potential for modern microbial and community genomics.

## INTRODUCTION

Modern high-throughput sequencing technologies provide comprehensive, large-scale datasets that have enabled a variety of novel genomic and metagenomic studies. A large number of statistical and computational tools have been developed specifically to tackle the complexity and high-dimensionality of such datasets and to provide robust and interpretable results. Visualizing data including thousands of microbial genomes or metagenomes, however, remains a challenging task that is often crucial to driving exploratory data mining and to compactly summarizing quantitative conclusions.

In the specific context of microbial genomics and metagenomics, next-generation sequencing in particular produces datasets of unprecedented size, including thousands of newly sequenced microbial genomes per month and a tremendous increase in genetic diversity sampled by isolates or culture-free assays. Displaying phylogenies with thousands of microbial taxa in hundreds of samples is infeasible with most available tools. This is especially true when sequencing profiles need to be placed in the context of sample

Corresponding author
Nicola Segata,
nicola.segata@unitn.it

metadata (e.g., clinical information). Among recently developed tools, iTOL (*Letunic & Bork, 2007*; *Letunic & Bork, 2011*) targets interactive analyses of large-scale phylogenies with a moderate amount of overlaid metadata, whereas ETE (*Huerta-Cepas, Dopazo & Gabaldon, 2010*) is a Python programming toolkit focusing on tree exploration and visualization that is targeted for scientific programmers, and Krona (*Ondov, Bergman & Phillippy, 2011*) emphasizes hierarchical quantitative information typically derived from metagenomic taxonomic profiles. Neither of these tools provides an automatable environment for non-computationally expert users in which very large phylogenies can be combined with high-dimensional metadata such as microbial community abundances, host or environmental phenotypes, or microbial physiological properties.

In particular, a successful high-throughput genomic visualization environment for modern microbial informatics must satisfy two criteria. First, software releases must be free and open-source to allow other researchers to verify and to adapt the software to their specific needs and to cope with the quick evolution of data types and datasets size. Second, visualization tools must be command-driven in order to be embedded in computational pipelines. This allows for a higher degree of analysis reproducibility, but the software must correspondingly be available for local installation and callable through a convenient interface (e.g., API or general scripting language). Local installations have also the advantage of avoiding the transfer of large or sensitive data to remote servers, preventing potential issues with the confidentiality of unpublished biological data. Neither of these criteria, of course, prevent tools from also being embeddable in web-based interfaces in order to facilitate use by users with limited computational expertise (*Blankenberg et al., 2010*; *Giardine et al., 2005*; *Goecks et al., 2010*; *Oinn et al., 2004*), and all such tools must regardless produce informative, clear, detailed, and publication-ready visualizations.

## MATERIALS & METHODS

GraPhlAn is a new tool for compact and publication-quality representation of circular taxonomic and phylogenetic trees with potentially rich sets of associated metadata. It was developed primarily for microbial genomic and microbiome-related studies in which the complex phylogenetic/taxonomic structure of microbial communities needs to be complemented with quantitative and qualitative sample-associated metadata. GraPhlAn is available at http://segatalab.cibio.unitn.it/tools/graphlan

### Implementation strategy

GraPhlAn is composed by two Python modules: one for drawing the image and one for adding annotations to the tree. GraPhlAn exploits the annotation file to highlight and personalize the appearance of the tree and of the associated information. The annotation file does not perform any modifications to the structure of the tree, but it just changes the way in which nodes and branches are displayed. Internally, GraPhlAn uses the matplotlib library (*Hunter, 2007*) to perform the drawing functions.

## The export2graphlan module

Export2graphlan is a framework to easily integrate GraPhlAn into already existing bioinformatics pipelines. Export2graphlan makes use of two external libraries: the pandas python library (*McKinney, 2012*) and the BIOM library, only when BIOM files are given as input.

Export2graphlan can take as input two files: the result of the analysis of MetaPhlAn (either version 1 or 2) or HUMAnN, and the result of the analysis of LEfSe. At least one of these two input files is mandatory. Export2graphlan will then produce a tree file and an annotation file that can be used with GraPhlAn. In addition, export2graphlan can take as input a BIOM file (either version 1 or 2).

Export2graphlan performs an analysis on the abundance values and, if present, on the LDA score assigned by LEfSe, to annotate and highlight the most abundant clades and the ones found to be biomarkers. Through a number of parameters the user can control the annotations produced by export2graphlan.

# RESULTS AND DISCUSSION

## Plotting taxonomic trees with clade annotations

The simplest structures visualizable by GraPhlAn include taxonomic trees (i.e., those without variable branch lengths) with simple clade or taxon nomenclature labels. These can be combined with quantitative information such as taxon abundances, phenotypes, or genomic properties. GraPhlAn provides separate visualization options for trees (thus potentially unannotated) and their annotations, the latter of which (the annotation module) attaches metadata properties using the PhyloXML format (*Han & Zmasek, 2009*). This annotation and subsequent metadata visualization process (Fig. 1) can be repeatedly applied to the same tree.

The GraPhlAn tree visualization (plotting module) takes as input a tree represented in any one of the most common data formats: Newick, Nexus (*Maddison, Swofford & Maddison, 1997*), PhyloXML (*Han & Zmasek, 2009*), or plain text. Without annotations, the plotting module generates a simple version of the tree (Fig. 1A), but the process can then continue by adding a diverse set of visualization annotations. Annotations can affect the appearance of the tree at different levels, including its global appearance ("global options" e.g., the size of the image, Fig. 1B), the properties of subsets of nodes and branches ("node options" e.g., the color of a taxon, Fig. 1C), and the background features used to highlight sub-trees ("label options" e.g., the name of a species containing multiple taxa, Fig. 1D). A subset of the available configurable options includes the thickness of tree branches, their colors, highlighting background colors and labels of specific sub-trees, and the sizes and shapes of individual nodes. Wild cards are supported to share graphical and annotation details among sub-trees by affecting all the descendants of a clade or its terminal nodes only. These features in combination aim to conveniently highlight specific sub-trees and metadata patterns of interest.

Additional taxon-specific features can be plotted as so-called external rings when not directly embedded into the tree. External rings are drawn just outside the area of the tree

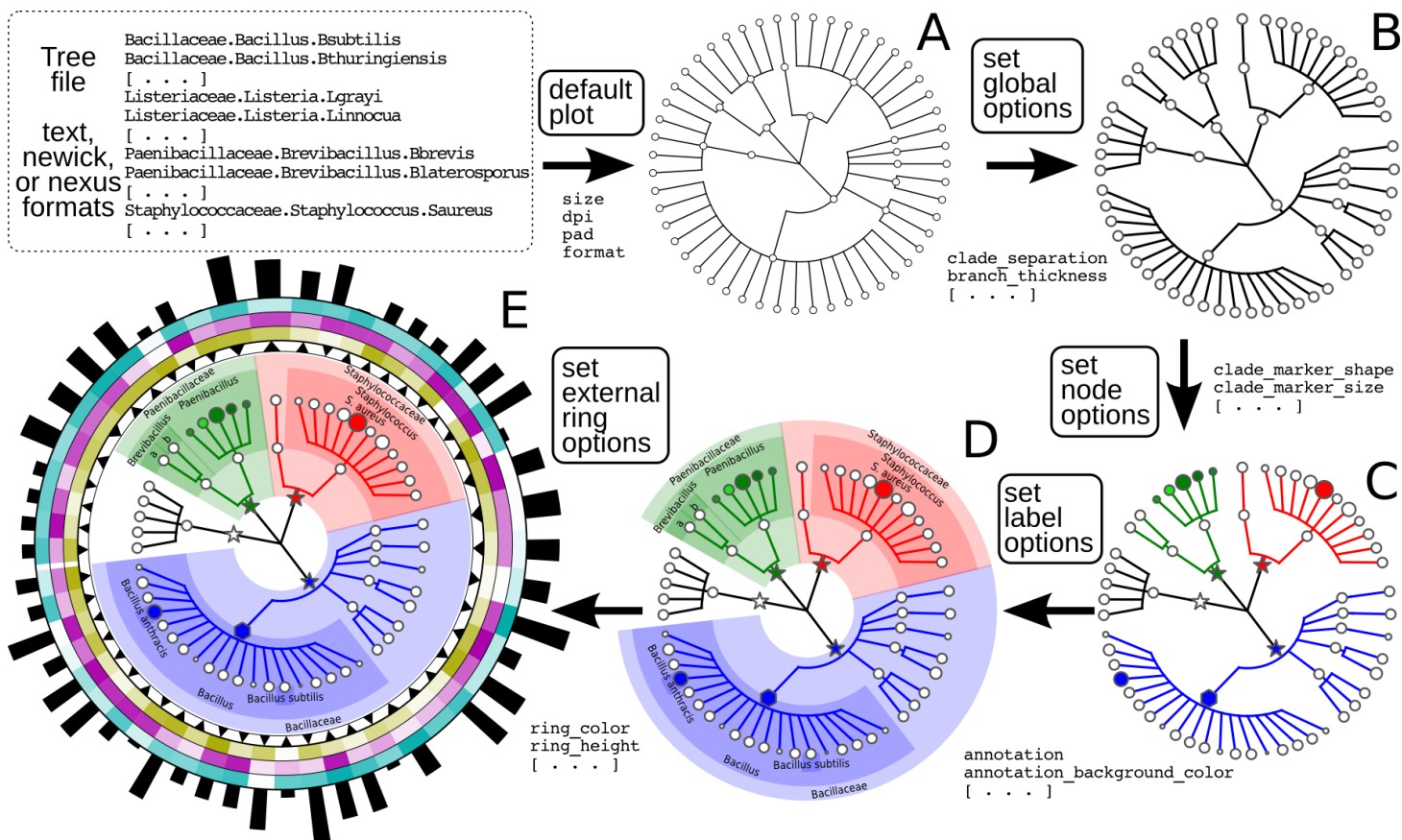

**Figure 1 Schematic and simplified example of GraPhlAn visualization of annotated phylogenies and taxonomies.** The software can start from a tree in Newick, Nexus, PhyloXML, or plain text formats. The "default plot" (A) produces a basic visualization of the tree's hierarchical structure. Through an annotation file, it is possible to configure a number of options that affect the appearance of the tree. For instance, some global parameters will affect the whole tree structure, such as the color and thickness of branches ("set global options," B). The same annotation file can act on specific nodes, customizing their shape, size, and color ("set node options," C). Labels and background colors for specific branches in the tree can also be configured ("set label options," D). External to the circular area of the tree, the annotation file can include directives for plotting different shapes, heatmap colors, or bar-plots representing quantitative taxon traits ("set external ring options," E).

and can be used to display specific information about leaf taxa, such as abundances of each species in different conditions/environments or their genome sizes. The shapes and forms of these rings are also configurable; for example, in Fig. 1E ("set external ring options"), the elements of the innermost external ring are triangular, indicating the directional sign of a genomic property. The second, third, and fourth external rings show leaf-specific features, using a heatmap gradient from blank to full color. Finally, the last external ring is a bar-plot representing a continuous property of leaf nodes of the tree.

## Compact representations of phylogenetic trees with associated metadata

Visualizing phylogenetic structures and their relation to external metadata is particularly challenging when the dimension of the internal structure is large. Mainly as a consequence of the low cost of sequencing, current research in microbial genomics and metagenomics
needs indeed to visualize a considerable amount of phylogenetic data. GraPhlAn can easily handle such cases, as illustrated here in an example of a large phylogenetic tree (3,737 taxa, provided as a PhyloXML file in the software repository, see Availability section) with multiple types of associated metadata (Fig. 2).

Specifically, we used GraPhlAn to display the microbial tree of life as inferred by PhyloPhlAn (*Segata et al., 2013*), annotating this evolutionary information with genome-specific metadata (Fig. 2). In particular, we annotated the genome contents related to seven functional modules from the KEGG database (*Kanehisa et al., 2012*), specifically two different ATP synthesis machineries (M00157: F-type ATPase and M00159: V/A-type ATPase) and five modules for bacterial fatty acid metabolism (M00082: Fatty acid biosynthesis, initiation, M00083: Fatty acid biosynthesis elongation, M00086: acyl-CoA synthesis, M00087: beta-Oxidation, and M00088: Ketone body biosynthesis). We then also annotated genome size as an external circular bar plot.

As expected, it is immediately visually apparent that the two types of ATPase are almost mutually exclusive within available genome annotations, with the V/A-type ATPase (module M00159) present mainly in *Archaea* and the F-type ATPase (module M000157) mostly characterizing *Bacteria*. Some exceptions are easily identifiable: *Thermi* and *Clamydophilia*, for instance, completely lack the F-type ATPase, presenting only the typically archaea-specific V/A-type ATPase. As previously discussed in the literature (*Cross & Müller, 2004*; *Mulkidjanian et al., 2007*), this may due to the acquisition of V/A-type ATPase by horizontal gene transfer and the subsequent loss of the F-type ATPase capability. Interestingly, some species such as those in the *Streptococcus* genus and some *Clostridia* still show both ATPase systems in their genomes.

With respect to fatty acid metabolism, some clades—including organisms such as *Mycoplasmas*—completely lack any of the targeted pathways. Indeed, *Mycoplasmas* are the smallest living cells yet discovered, lacking a cell wall (*Razin, 1992*) and demonstrating an obligate parasitic lifestyle. Since they primarily exploit host molecular capabilities, *Mycoplasmas* do not need to be able to fulfill all typical cell functions, and this is also indicated by the plotted very short genome sizes. *Escherichia*, on the other hand, has a much longer genome, and all the considered fatty acid metabolism capabilities are present. These evolutionary aspects are well known in the literature, GraPhlAn permits them and other phylogeny-wide genomic patterns to be easily visualized for further hypothesis generation.

## Visualizing microbiome biomarkers

GraPhlAn provides a means for displaying either phylogenetic (trees with branch lengths) or taxonomic (trees without branch length) data generated by other metagenomic analysis tools. For instance, we show here examples of GraPhlAn plots for taxonomic profiles (Fig. 3), functional profiles (Fig. 4), and specific features identified as biomarkers (Figs. 3 and 4). In these plots, GraPhlAn highlights microbial sub-trees that are found to be significantly differentially abundant by LEfSe (*Segata et al., 2011*), along with their effect sizes as estimated by linear discriminant analysis (LDA). To enhance biomarker

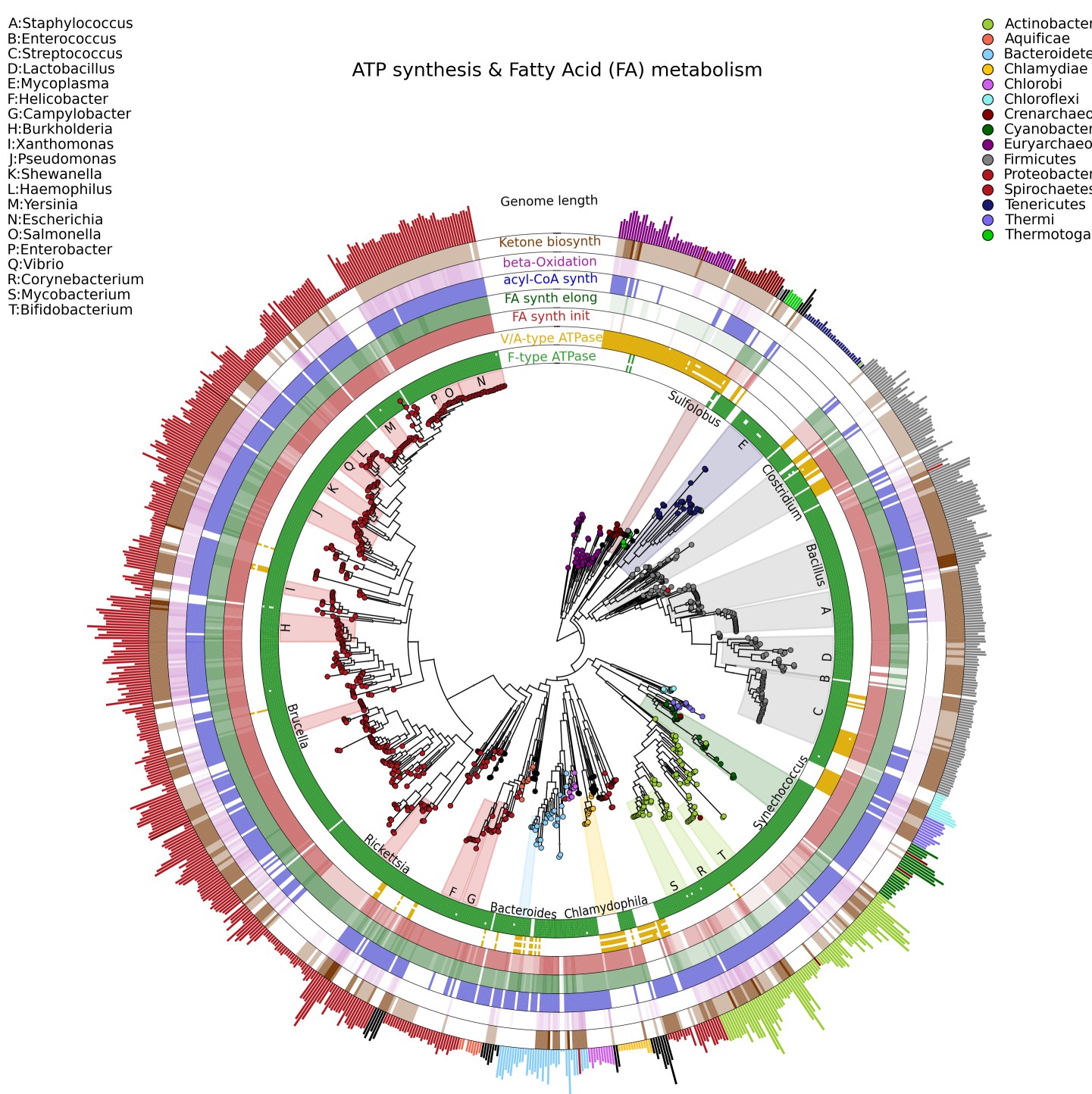

**Figure 2  A large, 3,737 genome phylogeny annotated with functional genomic properties.** We used the phylogenetic tree built using PhyloPhlAn (*Segata et al., 2013*) on all available microbial genomes as of 2013 and annotated the presence of ATP synthesis and Fatty Acid metabolism functional modules (as annotated in KEGG) and the genome length for all genomes. Colors and background annotation highlight bacterial phyla, and the functional information is reported in external rings. ATP synthesis rings visualize the presence (or absence) of each module, while Fatty Acid metabolism capability is represented with a gradient color. Data used in this image are available as indicated in the "Datasets used" paragraph, under "Materials and Methods" section.

A:Faecalibacterium
B:Faecalibacterium prausnitzii
C:Subdoligranulum
D:Subdoligranulum unclassified
E:Ruminococcus lactaris
F:Ruminococcus sp 5 1 39BFAA
G:Ruminococcus torques
H:Coprococcus
I:Coprococcus sp ART55 1
J:Coprococcus comes
K:Butyrivibrio
L:Butyrivibrio crossotus
M:Lachnospiraceae noname
N:Dorea longicatena
O:Roseburia hominis
P:Roseburia inulinivorans
Q:Eubacterium hallii
R:Eubacterium siraeum
S:Eubacterium eligens
T:Clostridium sp L2 50
U:Oscillibacter unclassified
V:Erysipelotrichaceae
W:Acidaminococcaceae
X:Alistipes putredinis
Y:Paraprevotella unclassified
Z:Bacteroides coprocola
a:Bacteroides caccae
b:Bacteroides uniformis
c:Bacteroides stercoris
d:Bacteroides eggerthii
e:Bacteroides sp 2 1 22
f:Bacteroides ovatus
g:Bacteroides thetaiotaomicron
h:Bacteroides vulgatus
i:Bacteroides xylanisolvens
j:Bacteroides plebeius
k:Barnesiella
l:Barnesiella intestinihominis
m:Parabacteroides unclassified
n:Parabacteroides merdae
o:Coriobacteriaceae
p:Bifidobacteriaceae
q:Bifidobacterium longum
r:Bifidobacterium adolescentis

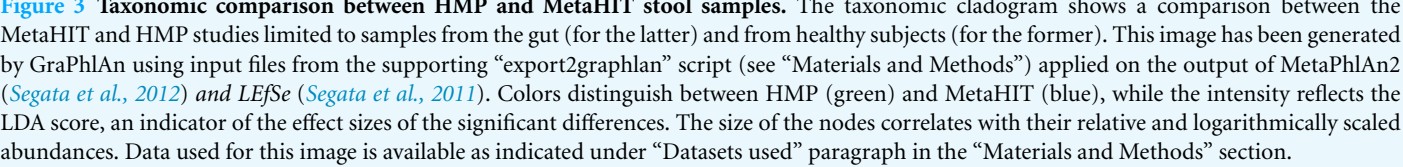

**Figure 3 Taxonomic comparison between HMP and MetaHIT stool samples.** The taxonomic cladogram shows a comparison between the MetaHIT and HMP studies limited to samples from the gut (for the latter) and from healthy subjects (for the former). This image has been generated by GraPhlAn using input files from the supporting "export2graphlan" script (see "Materials and Methods") applied on the output of MetaPhlAn2 (*Segata et al., 2012*) *and LEfSe* (*Segata et al., 2011*). Colors distinguish between HMP (green) and MetaHIT (blue), while the intensity reflects the LDA score, an indicator of the effect sizes of the significant differences. The size of the nodes correlates with their relative and logarithmically scaled abundances. Data used for this image is available as indicated under "Datasets used" paragraph in the "Materials and Methods" section.

A:Ribosome
B:Sulfur relay system
C:Protein export
D:Homologous recombination
E:Mismatch repair
F:Base excision repair
G:DNA replication
H:Starch and sucrose metabolism
I:Pentose and glucuronate interconversions
J:Pentose phosphate pathway
K:Glycolysis Gluconeogenesis
L:Citrate cycle TCA cycle
M:Drug metabolism other enzymes
N:Pyrimidine metabolism
O:Lipopolysaccharide biosynthesis
P:Peptidoglycan biosynthesis
Q:Lipoic acid metabolism
R:Vitamin B6 metabolism
S:Folate biosynthesis
T:Biotin metabolism
U:Nicotinate and nicotinamide metabolism
V:Thiamine metabolism
W:One carbon pool by folate
X:Pantothenate and CoA biosynthesis
Y:Terpenoid backbone biosynthesis
Z:Streptomycin biosynthesis
a:Sulfur metabolism
b:Carbon fixation in photosynthetic organisms
c:Cysteine and methionine metabolism
d:Phenylalanine tyrosine and tryptophan biosynthesis
e:Histidine metabolism
f:Lysine biosynthesis
g:Valine leucine and isoleucine biosynthesis
h:Arginine and proline metabolism
i:Alanine aspartate and glutamate metabolism
j:Selenocompound metabolism
k:D Glutamine and D glutamate metabolism
l:D Alanine metabolism
m:Bacterial secretion system
n:Phosphotransferase system PTS

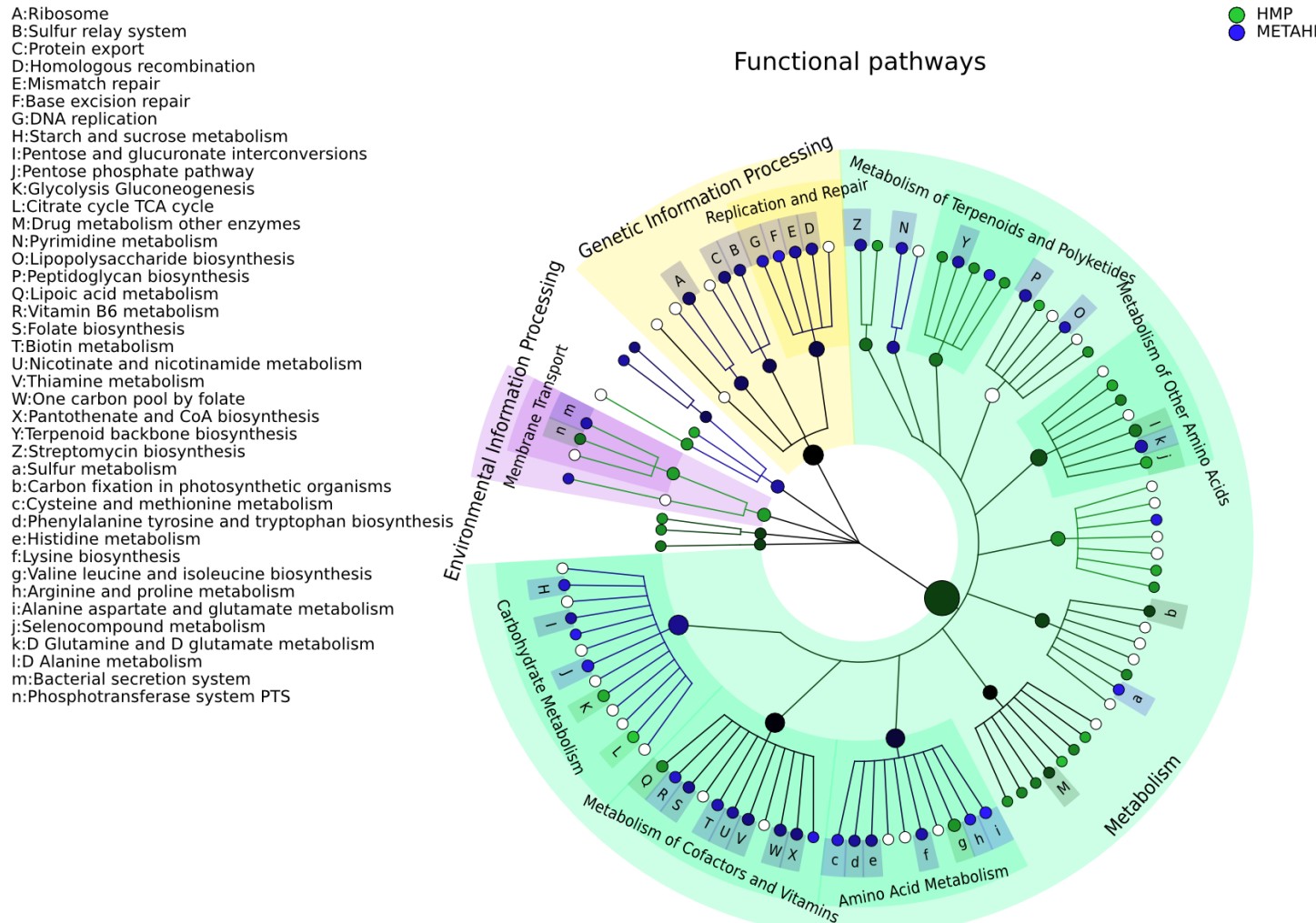

**Figure 4  Comparison of microbial community metabolic pathway abundances between HMP and MetaHIT.** Comparison of functional pathway abundances from the HMP (green) and MetaHIT (blue). This is the functional counterpart of the plot in Fig. 3 and was obtained applying GraPhlAn on HUMAnN (*Abubucker et al., 2012*) metabolic profiling. The intensity of the color represents the LDA score, and the sizes of the nodes are proportional to the pathway relative abundance estimated by HUMAnN. Three major groups are automatically highlighted by specifying them to the export2graphlan script: Environmental Information Processing, Genetic Information Processing, and Metabolism. Data used for this image is available as indicated under "Datasets used" paragraph in "Materials and Methods" section.

visualization, we annotated them in the tree with a shaded background color and with clade names as labels, with decreasing font sizes for internal levels. To represent the effect size, we scaled the node color from black (low LDA score) to full color (high LDA score).

Figure 3 shows the taxonomic tree of biomarkers (significantly differential clades) resulting from a contrast gut metagenome profiles from the Human Microbiome Project (HMP) (*Huttenhower et al., 2012*) and MetaHIT samples (*Qin et al., 2010*). Only samples from healthy individuals in the latter cohort were included. The filtered dataset was analyzed using LEfSe (*Segata et al., 2011*) and the cladogram obtained using the *export2graphlan* script provided with GraPhlAn and discussed in the following section. As expected, the image highlights that *Firmicutes* and *Bacteroides* are the two most abundant

taxa in the healthy gut microbiome (*David et al., 2014*; *Wu et al., 2011*). The *Bacteroidetes* phylum contains many clades enriched in the HMP dataset, while *Firmicutes* show higher abundances for MetaHIT samples. GraPhlAn can thus serve as a visual tool for inspecting specific significant differences between conditions or cohorts.

Functional ontologies can be represented by GraPhlAn in a similar way and provide complementary features to the types of taxonomic analyses shown above. Metabolic profiles quantified by HUMAnN (*Abubucker et al., 2012*) using KEGG (*Kanehisa et al., 2014*) from the same set of HMP and MetaHIT samples are again contrasted on multiple functional levels in Fig. 4. The tree highlights three different broad sets of metabolic pathways: Environmental Information Processing, Genetic Information Processing, and Metabolism, with the last being the largest subtree. More specific metabolic functions are specifically enriched in the HMP cohort, such as Glycolysis and the Citrate cycle, or in the MetaHIT cohort, such as Sulfur Metabolism and Vitamin B6 Metabolism. This illustrates GraPhlAn's use with different types of data, such as functional trees in addition to taxonomies or phylogenies. By properly configuring input parameters of *export2graphlan*, we automatically obtained both Figs. 3 and 4 (bash scripts used for these operations are available in the GraPhlAn software repository).

## Reproducible integration with existing analysis tools and pipelines

Graphical representations are usually a near- final step in the complex computational and metagenomic pipelines, and automating their production is crucial for convenient but reproducible analyses. To this end, GraPhlAn has been developed with command-driven automation in mind, as well as flexibility in the input "annotation file" so as to be easily generated by automated scripts. Depending on the specific analysis, these scripts can focus on a diverse set of commands to highlight the features of interest. Despite this flexibility, we further tried to ease the integration of GraPhlAn by providing automatic offline conversions for some of the available metagenomic pipelines and by embedding it into the well-established Galaxy web framework (*Blankenberg et al., 2010*; *Giardine et al., 2005*; *Goecks et al., 2010*).

In order to automatically generate GraPhlAn plots from a subset of available shotgun metagenomic tools comprising MetaPhlAn (for taxonomic profiling), HUMAnN (for metabolic profiling), and LEfSe (for biomarker discovery), we developed a script named "export2graphlan" able to convert the outputs of these tools into GraPhlAn input files as schematized in Fig. 5. This conversion software is also meant to help biologists by providing initial, automated input files for GraPhlAn that can then be manually tweaked for specific needs such as highlighting clades of particular interest. The export2graphlan framework can further accept the widely adopted BIOM format, both versions 1 and 2 (*McDonald et al., 2012*). This makes it possible to readily produce GraPhlAn outputs from other frameworks such as QIIME (*Caporaso et al., 2010*) and mothur (*Schloss et al., 2009*) for 16S rRNA sequencing studies.

A web-based deployment of the GraPhlAn application is available to the public via Galaxy at http://huttenhower.sph.harvard.edu/galaxy/. The Galaxy interface of GraPhlAn

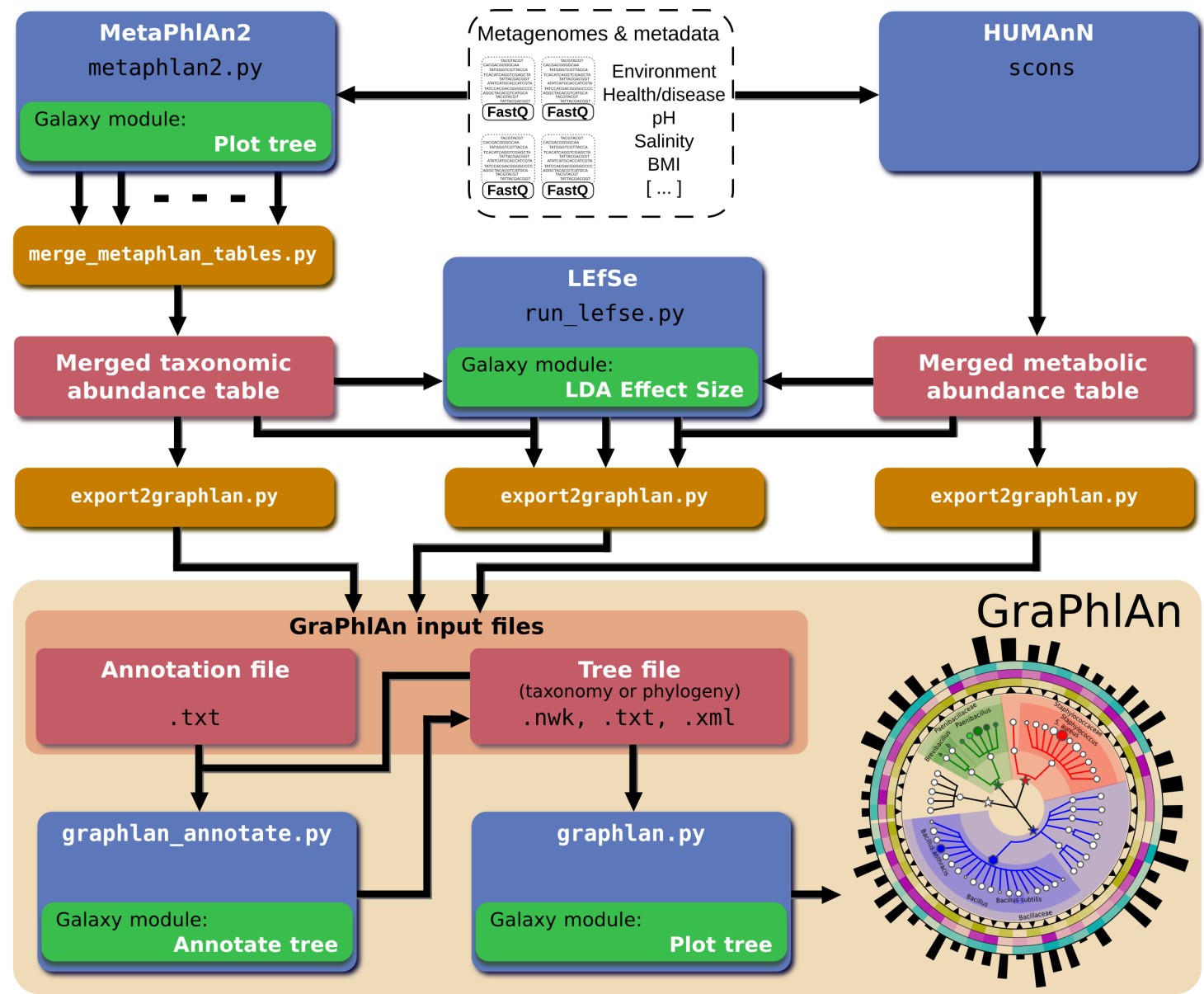

**Figure 5 Integration of GraPhlAn into existing analyses pipelines.** We developed a conversion framework called "export2graphlan" that can deal with several output formats from different analysis pipelines, generating the necessary input files for GraPhlAn. Export2graphlan directly supports MetaPhlAn2, LEfSe, and HUMAnN output files. In addition, it can also accept BIOM files (both version 1 and 2), making GraPhlAn available for tools supporting this format including the QIIME and mothur systems. The tools can be ran on local machine as well as through the Galaxy web system using the modules reported in green boxes.

consists of four processing modules: (1) *Upload file*, that manages the upload of the input data into Galaxy; (2) *GraPhlAn Annotate Tree*, which allows the user to specify the annotations that will be applied to the final image; (3) *Add Rings to tree*, an optional step to select an already uploaded file in Galaxy that will be used as an annotation file for the external rings; and (4) *Plot tree*, that sets some image parameters such as the size, the resolution, and the output format.

## CONCLUSIONS

We present GraPhlAn, a new method for generating high-quality circular phylogenies potentially integrated with diverse, high-dimensional metadata. We provided several examples showing the application of GraPhlAn to phylogenetic, functional, and taxonomic summaries. The system has already been used for a variety of additional visualization tasks, including highlighting the taxonomic origins of metagenomic biomarkers (*Segata et al., 2012*; *Segata et al., 2011*; *Shogan et al., 2014*; *Xu et al., 2014*), exposing specific microbiome metabolic enrichments within a functional ontology (*Abubucker et al., 2012*; *Sczesnak et al., 2011*), and representing 16S rRNA sequencing results (*Ramirez et al., 2014*). However, GraPhlAn is not limited to microbiome data and has additionally been applied to animal and plant taxonomies (*Tree of Sex Consortium, 2014*) and to large prokaryotic phylogenies built using reference genomes (*Baldini et al., 2014*; *Chai et al., 2014*; *Langille et al., 2013*; *Segata et al., 2013*).

Compared to the other existing state-of-the-art approaches such as Krona (*Ondov, Bergman & Phillippy, 2011*) and iTOL (*Letunic & Bork, 2007*; *Letunic & Bork, 2011*), GraPhlAn provides greater flexibility, configuration, customization, and automation for publication reproducibility. It is both easily integrable into automated computational pipelines and can be used conveniently online through the Galaxy-based web interface. The software is available open-source, and the features highlighted here illustrate a number of ways in which its visualization capabilities can be integrated into microbial and community genomics to display large tree structures and corresponding metadata.

## DATA AND SOFTWARE AVAILABILITY

### Description of the datasets and figure generation

The data of the taxonomic trees presented in Fig. 1 is available in the *guide* folder, inside the *examples* directory of the GraPhlAn repository (https://bitbucket.org/nsegata/graphlan). This same image is thoroughly described under the "A step-by-step example" section, in the GraPhlAn wiki included in the repository.

The genomic data used for the Tree of Life in Fig. 2 was obtained from the Integrated Microbial Genomes (IMG) data management system of the U.S. Department of Energy Joint Genome Institute (DOE JGI) 2.0 dataset (http://jgi.doe.gov/news_12_1_06/). From the KEGG database (*Kanehisa & Goto, 2000*; *Kanehisa et al., 2014*) we focused on the following modules: M00082, M00083, M00086, M00087, M00088, M00157, and M00159. The input data for drawing Fig. 2 is available in the *PhyloPhlAn* folder under the *examples* directory of the GraPhlAn repository.

In Fig. 3, to comprehensively characterize the asymptomatic human gut microbiota, we combined 224 fecal samples (>17 million reads) from the Human Microbiome Project (HMP) (*Human Microbiome Project C, 2012a*; *Human Microbiome Project C, 2012b*) and the MetaHIT (*Qin et al., 2010*) projects, two of the largest gut metagenomic collections available. The taxonomic profiles were obtained by applying MetaPhlAn2. The 139 fecal samples from the HMP can be accessed at http://hmpdacc.org/HMASM/, whereas the 85 fecal samples from MetaHIT were downloaded from the European Nucleotide

Archive (http://www.ebi.ac.uk/ena/, study accession number ERP000108). The input files for obtaining this image with GraPhlAn are present into the *examples* folder of the repository, inside the *hmp_metahit* directory. The two input files represent the merge result of the MetaPhlAn analysis (*hmp_metahit.txt*) and the LEfSe result on the first file (*hmp_metahit.lefse.txt*). The bash script provided exploits the export2graphlan capabilities to generate the annotation file.

The functional profiles used in Fig. 4 are the reconstruction of the metabolic activities of microbiome communities. The HUMAnN pipeline (*Abubucker et al., 2012*) infers community function directly from short metagenomic reads, using the KEGG ortholog (KO) groups. HUMAnN was run on the same samples of Fig. 3. The dataset is available online at http://www.hmpdacc.org/HMMRC/. As for the previous figure, the input files for obtaining Fig. 4 are uploaded in the *hmp_metahit_functional* folder, inside the *examples* directory of the repository. The two files (*hmp_metahit_functional.txt* and *hmp_metahit_functional.lefse.txt*) represent the result of HUMAnN on the HMP and MetaHIT datasets and the result of LEfSe executed on the former file. The bash script provided executes export2graphlan for generating the annotation file and then invoking GraPhlAn for plotting the functional tree.

The dataset of Fig. S1 refer to a 16S rRNA amplicon experiment. Specifically, it consists of 454 FLX Titanium sequences spanning the V3 to V5 variable regions, obtained from 24 healthy samples (12 male and 12 female) for a total of 301 samples. Detailed protocols used for enrollment, sampling, DNA extraction, 16S amplification and sequencing are available on the Human Microbiome Project Data Analysis and Coordination Center website HMP Data Analysis and Coordination Center (http://www.hmpdacc.org/tools_protocols/tools_protocols.php). This data are pilot samples from the HMP project (*Segata et al., 2011*). The input files for obtaining this image is available in the *examples* folder of the export2graphlan repository (https://bitbucket.org/CibioCM/export2graphlan), inside the *hmp_aerobiosis* directory. The two files represent the taxonomic tree of the HMP project and the results of LEfSe executed on the same data.

In Fig. S2 we used the saliva microbiome profiles obtained by 16S rRNA sequencing on the IonTorrent platform (amplifying the hypervariable region V3). The dataset comprises a total of 13 saliva samples from healthy subjects as described in (*Dassi et al., 2014*) and it is available in the NCBI Short Read Archive (http://www.ncbi.nlm.nih.gov/sra). The input BIOM file for drawing this image is available in the *saliva_microbiome* directory inside the *examples* folder of the GraPhlAn repository.

For Fig. S3 data represent the temporal dynamics of the human vaginal microbiota, and were taken from the study of (*Gajer et al., 2012*). Data were obtained by 16S rRNA using the 454 pyrosequencing technology (sequencing the V1 and V2 hypervariable regions). The dataset is composed of samples from 32 women that self-collected samples twice a week for 16 weeks. The input file, provided in BIOM format, is present in the *vaginal_microbiota* folder inside the *examples* directory of the GraPhlAn repository.

### Software repository, dependences, and user support

GraPhlAn is freely available (http://segatalab.cibio.unitn.it/tools/graphlan) and released open-source in Bitbucket (https://bitbucket.org/nsegata/graphlan) with a set of working examples and a complete tutorial that guides users throughout its functionality. GraPhlAn uses the matplotlib library (*Hunter, 2007*). GraPhlAn is also available via a public Galaxy instance at http://huttenhower.sph.harvard.edu/galaxy/.

Export2graphlan is freely available and released open-source in Bitbucket (https://bitbucket.org/CibioCM/export2graphlan) along with a number of examples helpful for testing if everything is correctly configured and installed. The export2graphlan repository is also present as a sub-repository inside the GraPhlAn repository. The export2graphlan module exploits the pandas library (*McKinney, 2012*) and the BIOM library (*McDonald et al., 2012*).

Both GraPhlAn and export2graphlan are supported through the Google group "GraPhlAn-users" (https://groups.google.com/forum/#!forum/graphlan-users), available also as a mailing list at: graphlan-users@googlegroups.com.

## ACKNOWLEDGEMENTS

We would like to thank the members of the Segata and Huttenhower labs for helpful suggestions, the WebValley team and participants for inspiring comments and tests, and the users that tried the alpha version of GraPhlAn providing invaluable feedback to improve the software.

### Funding

This work has been supported by NIH R01HG005969, NSF CAREER DBI-1053486, and W911NF-11-1-0473 to C.H. and by the People Programme (Marie Curie Actions) of the European Union's Seventh Framework Programme (FP7/2007-2013) under REA grant agreement no PCIG13-GA-2013-618833, startup funds from the Centre for Integrative Biology (University of Trento), by MIUR "Futuro in Ricerca" E68C13000500001, by Terme di Comano, and by from Fondazione Caritro to N.S. The funders had no role in study design, data collection and analysis, decision to publish, or preparation of the manuscript.

### Grant Disclosures

The following grant information was disclosed by the authors:
NIH: R01HG005969.
NSF CAREER: DBI-1053486.
W911NF-11-1-0473.
European Union's Seventh Framework Programme: PCIG13-GA-2013-618833.
MIUR: E68C13000500001.

### Competing Interests

The authors declare there are no competing interests.

## Author Contributions

- Francesco Asnicar performed the experiments, analyzed the data, contributed reagents/materials/analysis tools, wrote the paper, prepared figures and/or tables, reviewed drafts of the paper.
- George Weingart conceived and designed the experiments, contributed reagents/materials/analysis tools, reviewed drafts of the paper.
- Timothy L. Tickle contributed reagents/materials/analysis tools, reviewed drafts of the paper.
- Curtis Huttenhower conceived and designed the experiments, contributed reagents/materials/analysis tools, wrote the paper, reviewed drafts of the paper.
- Nicola Segata conceived and designed the experiments, performed the experiments, analyzed the data, contributed reagents/materials/analysis tools, wrote the paper, reviewed drafts of the paper.

## Data Deposition

The following information was supplied regarding the deposition of related data:
https://bitbucket.org/nsegata/graphlan.

## Supplemental Information

Supplemental information for this article can be found online at http://dx.doi.org/10.7717/peerj.1029#supplemental-information.

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
