# Peer review of "Compact graphical representation of phylogenetic data and metadata with GraPhlAn"

_PeerJ, doi:10.7717/peerj.1029_

## Round 0.1 · original submission · Minor Revisions

Please address the comments of the reviewers, especially about providing enough information to be able to reproduce the figures of the paper.

Reviewer 1 ·

Basic reporting

In the work presented by Asnicar et al, entitled “Compact graphical representation of phylogenetic data and metadata with GraPhlAn”, the authors present a software tool that allows for graphical representation of metagenetic and metagenomic hierarchichal information and the associated metadata using circular phylogeny structures.
While GraPhlAn does not represent and algorithmic advance or an implementation of analysis software with improved performance/features, it is useful as a readily usable tool for graphical representation, based on well-known standard formats and tools and perfectly fits into the metagenomics research field.
The article is well written, self-contained and appropriate figures are displayed to illustrate examples that range from simple to complex use of GraPhlAn tool using well-known public datasets.

Experimental design

Since the manuscript deals with a software tool there is no experimental design as such. However, authors report a methods sections that, along with widely used code repository, describe software implementation strategy adequately allowing for code examination and modification.

Validity of the findings

GraPhlAn has been implemented in both stand-alone and web-server formats, is nowadays widely used in the metagenomics research field and has been used for graphical representation in several microbiome based studies. This software tool adds to several other tools for graphical representation, such as the ones cited in the manuscript, and further facilitates visualization and hypothesis generation using metagenomics data.
Furthermore, GraPhlAn software tool is open-source, well documented and supported by the community.

·

Basic reporting

The article is clearly written and lovely figures (which, given the subject, I suppose we should expect). I would have expected to see references to more of the existing literature on phylogeny visualization, specifically ETE, http://etetoolkit.org/, which shares many features with the software described here.

Experimental design

This paper describes a software package rather than a traditional research study. Source code is available in a public code repository with an open-source license, allowing for inspection and reuse.

Validity of the findings

The paper describes the data underlying each figure and the source database, but does not provide the actual datasets and GraPhlAn input files. For example, the data for Supplementary Figure 2 is "available in the NCBI Short Read Archive". It would be a fair bit of work to assemble the dataset and reproduce the figure, particuarly since the manuscript has very little detail about how to format metadata for display on trees.

There is an examples directory in the bitbucket repo [1] but these do not seem to correspond perfectly with the figures in the text.

[1] https://bitbucket.org/nsegata/graphlan/src/cea7562e151e410d65fbd2381ff75cec7364fad6/examples/?at=default

The source code repository has good documentation and tutorials, and the authors seem responsive to users on the Google group.

---

## Round 0.2 · accepted · Accept

The authors have addressed the reviewer's comments in a satisfactory way.

---

## Author Rebuttal · Round 0.2

CIBIO - Centre for Integrative Biology
Trento, Italy

Dear Editor,

Many thanks for your positive feedback and we greatly appreciate the comments from the reviewers that helped to improve and clarify the manuscript. Following your suggestions here are the changes we made on the manuscript:

- We completed the brief literature review related tools by adding the ETE toolkit Python programming environment, as suggested by one of the reviewers.

- We now provide complete scripts and step-by-step procedures to reproduce the figures presented in the manuscript. The scripts and supporting data files are included in the online GraPhlAn repository and linked from the manuscript.

We edited as per the points above, and a point-by-point response is provided below (text in green color) with references to the modified text (text in red color). We also slightly restructured the paper by moving the "Materials and Methods" section before the "Results and Discussion" section as recommended, leaving just the sections about availability and datasets used at the end.

Many thanks again for your feedback, we continue to be open to any suggestions you or the reviewers can provide to further finalize the manuscript.

Sincerely,

Dr. Nicola Segata

Principal Investigator
Centre for Integrative Biology
University of Trento (Italy)

E-mail: nicola.segata@unitn.it

Via Sommarive, 9 – 38123 Povo (Trento) - Italy

[Figure] [Figure]

CIBIO - Centre for Integrative Biology
Trento, Italy

## Replies to the points raised by the Reviewer #1:

### Basic reporting

In the work presented by Asnicar et al, entitled "Compact graphical representation of phylogenetic data and metadata with GraPhlAn", the authors present a software tool that allows for graphical representation of metagenetic and metagenomic hierarchichal information and the associated metadata using circular phylogeny structures.

While GraPhlAn does not represent and algorithmic advance or an implementation of analysis software with improved performance/features, it is useful as a readily usable tool for graphical representation, based on well-known standard formats and tools and perfectly fits into the metagenomics research field.

The article is well written, self-contained and appropriate figures are displayed to illustrate examples that range from simple to complex use of GraPhlAn tool using well-known public datasets.

### Experimental design

Since the manuscript deals with a software tool there is no experimental design as such. However, authors report a methods sections that, along with widely used code repository, describe software implementation strategy adequately allowing for code examination and modification.

### Validity of the findings

GraPhlAn has been implemented in both stand-alone and web-server formats, is nowadays widely used in the metagenomics research field and has been used for graphical representation in several microbiome based studies. This software tool adds to several other tools for graphical representation, such as the ones cited in the manuscript, and further facilitates visualization and hypothesis generation using metagenomics data.

Furthermore, GraPhlAn software tool is open-source, well documented and supported by the community.

We would like to thank the reviewer for the nice summary of our work and his/her positive feedback.

## Replies to the points raised by the Reviewer #2 (Karen Cranston):

### Basic reporting

The article is clearly written and lovely figures (which, given the subject, I suppose we should expect). I would have expected to see references to more of the existing literature on phylogeny visualization, specifically ETE, http://etetoolkit.org/, which shares many features with the software described here.

We indeed missed the ETE reference, and we thank the reviewer pointing this out. We now added ETE in the revised brief review of the state-of-the-art.

> In the specific context of microbial genomics and metagenomics, next-generation sequencing in particular produces datasets of unprecedented size, including thousands of newly sequenced microbial genomes per month and a tremendous increase in genetic diversity sampled by isolates or culture-free assays. Displaying phylogenies with thousands of microbial taxa in hundreds of samples is infeasible with most available tools. This is especially

true when sequencing profiles need to be placed in the context of sample metadata (e.g. clinical information). Among recently developed tools, iTOL (Letunic & Bork 2007; Letunic & Bork 2011) targets interactive analyses of large-scale phylogenies with a moderate amount of overlaid metadata, whereas ETE (Huerta-Cepas et al. 2010) is a Python programming toolkit focusing on tree exploration and visualization that is targeted for scientific programmers, and Krona (Ondov et al. 2011) emphasizes hierarchical quantitative information typically derived from metagenomic taxonomic profiles. Neither of these tools provides an automatable environment for non-computationally expert users in which very large phylogenies can be combined with high-dimensional metadata such as microbial community abundances, host or environmental phenotypes, or microbial physiological properties.

**Experimental design**

This paper describes a software package rather than a traditional research study. Source code is available in a public code repository with an open-source license, allowing for inspection and reuse.

**Validity of the findings**

The paper describes the data underlying each figure and the source database, but does not provide the actual datasets and GraPhlAn input files. For example, the data for Supplementary Figure 2 is "available in the NCBI Short Read Archive". It would be a fair bit of work to assemble the dataset and reproduce the figure, particulary since the manuscript has very little detail about how to format metadata for display on trees.

There is an examples directory in the bitbucket repo [1] but these do not seem to correspond perfectly with the figures in the text.

[1]
https://bitbucket.org/nsegata/graphlan/src/cea7562e151e410d65fbd2381ff75cec7364fad6/examples/?at=default

Our purpose was indeed to provide the full step-by-step guide for each figure, and we thank the reviewer pointing out some missing parts. We now added the scripts for all main figures and provide the links from the manuscript to the corresponding "example" folder in the public repository. In the case of the supplementary figure S2, it would take a bit of work to assemble the dataset, process it through QIIME and then get the BIOM resulting file. However, the purpose of the supplementary figure S2 is to show the export2graphlan capabilities of handle BIOM files, hence enabling GraPhlAn to draw the data produced within the QIIME system. For this case we thus decided to provide directly the BIOM file resulting from QIIME application and start the pipeline from there. We modified the text as reported below, also clearly stating which folder in the GraPhlAn repository contains the input and annotations file for generating the figures showed in the paper.

CIBIO - Centre for Integrative Biology
Trento, Italy

**Description of the datasets and figure generation**

The data of the taxonomic trees presented in **Fig. 1** is available in the *guide* folder, inside the *examples* directory of the GraPhlAn repository (https://bitbucket.org/nsegata/graphlan). This same image is thoroughly described under the "A step-by-step example" section, in the GraPhlAn wiki included in the repository.

The genomic data used for the Tree of Life in **Fig. 2** was obtained from the Integrated Microbial Genomes (IMG) data management system of the U.S. Department of Energy Joint Genome Institute (DOE JGI) 2.0 dataset (http://jgi.doe.gov/news_12_1_06/). From the KEGG database (Kanehisa & Goto 2000; Kanehisa et al. 2014) we focused on the following modules: M00082, M00083, M00086, M00087, M00088, M00157, and M00159. The input data for drawing **Fig. 2** is available in the *PhyloPhlAn* folder under the *examples* directory of the GraPhlAn repository.

In **Fig. 3**, to comprehensively characterize the asymptomatic human gut microbiota, we combined 224 fecal samples (>17 million reads) from the Human Microbiome Project (HMP) (Human Microbiome Project 2012a; Human Microbiome Project 2012b) and the MetaHIT (Qin et al. 2010) projects, two of the largest gut metagenomic collections available. The taxonomic profiles were obtained by applying MetaPhlAn2. The 139 fecal samples from the HMP can be accessed at http://hmpdacc.org/HMASM/, whereas the 85 fecal samples from MetaHIT were downloaded from the European Nucleotide Archive (http://www.ebi.ac.uk/ena/, study accession number ERP000108). The input files for obtaining this image with GraPhlAn are present into the *examples* folder of the repository, inside the *hmp_metahit* directory. The two input files represent the merge result of the MetaPhlAn analysis (*hmp_metahit.txt*) and the LEfSe result on the first file (*hmp_metahit.lefse.txt*). The bash script provided exploits the export2graphlan capabilities to generate the annotation file.

The functional profiles used in **Fig. 4** are the reconstruction of the metabolic activities of microbiome communities. The HUMAnN pipeline (Abubucker et al. 2012) infers community function directly from short metagenomic reads, using the KEGG ortholog (KO) groups. HUMAnN was run on the same samples of **Fig. 3**. The dataset is available on-line at http://www.hmpdacc.org/HMMRC/. As for the previous figure, the input files for obtaining **Fig. 4** are uploaded in the *hmp_metahit_functional* folder, inside the *examples* directory of the repository. The two files (*hmp_metahit_functional.txt* and *hmp_metahit_functional.lefse.txt*) represent the result of HUMAnN on the HMP and MetaHIT datasets and the result of LEfSe executed on the former file. The bash script provided executes export2graphlan for generating the annotation file and then invoking GraPhlAn for plotting the functional tree.

The dataset of supplementary **Fig. S1** refer to a 16S rRNA amplicon experiment. Specifically, it consists of 454 FLX Titanium sequences spanning the V3 to V5 variable regions, obtained from 24 healthy samples (12 male and 12 female) for a total of 301 samples. Detailed protocols used for enrollment, sampling, DNA extraction, 16S amplification and sequencing are available on the Human Microbiome Project Data Analysis and Coordination Center website HMP Data Analysis and Coordination Center (http://www.hmpdacc.org/tools_protocols/tools_protocols.php). This data are pilot samples from the HMP project (Segata et al. 2011). The input files for obtaining this image is available

in the *examples* folder of the export2graphlan repository (https://bitbucket.org/CibioCM/export2graphlan), inside the *hmp_aerobiosis* directory. The two files represent the taxonomic tree of the HMP project and the results of LEfSe executed on the same data.

In the supplementary **Fig. S2** we used the saliva microbiome profiles obtained by 16S rRNA sequencing on the IonTorrent platform (amplifying the hypervariable region V3). The dataset comprises a total of 13 saliva samples from healthy subjects as described in (Dassi et al. 2014) and it is available in the NCBI Short Read Archive (http://www.ncbi.nlm.nih.gov/sra). The input BIOM file for drawing this image is available in the *saliva_microbiome* directory inside the *examples* folder of the GraPhlAn repository.

For the supplementary **Fig. S3** data represent the temporal dynamics of the human vaginal microbiota, and were taken from the study of (Gajer et al. 2012). Data were obtained by 16S rRNA using the 454 pyrosequencing technology (sequencing the V1 and V2 hypervariable regions). The dataset is composed of samples from 32 women that self-collected samples twice a week for 16 weeks. The input file, provided in BIOM format, is present in the *vaginal_microbiota* folder inside the *examples* directory of the GraPhlAn repository.

The source code repository has good documentation and tutorials, and the authors seem responsive to users on the Google group.